First body of evidence suggesting a role of a tankyrase-binding motif (TBM) of vinculin (VCL) in epithelial cells

Vilchez Larrea Salomé 1
http://orcid.org/0000-0002-8066-5445 Valsecchi Wanda Mariela 2 3
Fernández Villamil Silvia H. 1 3 silvia.villamil@gmail.com
http://orcid.org/0000-0002-4239-2802 Lafon Hughes Laura I. 4 5 lauralafon2010@gmail.com
1 Instituto de Investigaciones en Ingeniería Genética y Biología Molecular “Dr Héctor N. Torres”, Consejo Nacional de Investigaciones Científicas y Técnicas , Buenos Aires, Ciudad Autónoma de Buenos Aires , República Argentina
2 Instituto de Química y Fisicoquímica Biológicas, “Prof. Alejandro C. Paladini” (IQUIFIB) Consejo Nacional de Investigaciones Científicas y Técnicas (CONICET) , Buenos Aires, Ciudad Autónoma de Buenos Aires , República Argentina
3 Departamento de Química Biológica, Facultad de Farmacia y Bioquímica, Universidad de Buenos Aires , Buenos Aires, Ciudad Autónoma de Buenos Aires , Argentina
4 Grupo de Biofisicoquímica, Departamento de Ciencias Biológicas, Centro Universitario Regional Litoral Norte (CENUR), Universidad de la República , Salto , Uruguay
5 Departamento de Genética, Instituto de Investigaciones Biológicas Clemente Estable, Ministerio de Educación y Cultura , Montevideo , Uruguay
Gould Gwyn
Electronic publication date: 2021 May 27
Publication date: 2021
Volume: 9
Electronic Location ID: e11442
Received 2020 Nov 23; Accepted 2021 Apr 21
Copyright: © 2021 Vilchez Larrea et al.
Copyright year: 2021
Copyright holder: Vilchez Larrea et al.
License: This is an open access article distributed under the terms of the Creative Commons Attribution License, which permits unrestricted use, distribution, reproduction and adaptation in any medium and for any purpose provided that it is properly attributed. For attribution, the original author(s), title, publication source (PeerJ) and either DOI or URL of the article must be cited.
License URL: https://creativecommons.org/licenses/by/4.0/

Keywords: Poly(ADP)ribose, PARP, Tankyrase, Epithelia, Adherens junctions, Vinculin, Vero, MCF-7, Tankyrase-binding motif (TBM), Epithelial to mesenchymal transition (EMT)

Funding: Consejo Nacional de Investigaciones Científicas y Técnicas (CONICET, Argentina) Agencia Nacional de Investigación e Innovación (ANII, Uruguay) MOV_CO_2015_1_110430 This work was partially supported by Consejo Nacional de Investigaciones Científicas y Técnicas (CONICET, Argentina) and Agencia Nacional de Investigación e Innovación (ANII, Uruguay) through a bilateral cooperation project (MOV_CO_2015_1_110430) and Programa de Desarrollo de las Ciencias Básicas (PEDECIBA, Uruguay). Also, the author Laura I Lafon Hughes donated US$ 1000 to Instituto Pasteur de Montevideo (IPMONT, 21/05/2019) to compensate for usage of required molecular biology reagents. The CRISPR/Cas VCL KO kit was donated by Synthego. The funders had no role in study design, data collection and analysis, decision to publish, or preparation of the manuscript.

==============================
Background

Adherens junctions (AJ) are involved in cancer, infections and neurodegeneration. Still, their composition has not been completely disclosed. Poly(ADP-ribose) polymerases (PARPs) catalyze the synthesis of poly(ADP-ribose) (PAR) as a posttranslational modification. Four PARPs synthesize PAR, namely PARP-1/2 and Tankyrase-1/2 (TNKS). In the epithelial belt, AJ are accompanied by a PAR belt and a subcortical F-actin ring. F-actin depolymerization alters the AJ and PAR belts while PARP inhibitors prevent the assembly of the AJ belt and cortical actin. We wondered which PARP synthesizes the belt and which is the PARylation target protein. Vinculin (VCL) participates in the anchorage of F-actin to the AJ, regulating its functions, and colocalized with the PAR belt. TNKS has been formerly involved in the assembly of epithelial cell junctions.

Hypothesis

TNKS poly(ADP-ribosylates) (PARylates) epithelial belt VCL, affecting its functions in AJ, including cell shape maintenance.

Materials and Methods

Tankyrase-binding motif (TBM) sequences in hVCL gene were identified and VCL sequences from various vertebrates, Drosophila melanogaster and Caenorhabditis elegans were aligned and compared. Plasma membrane-associated PAR was tested by immunocytofluorescence (ICF) and subcellular fractionation in Vero cells while TNKS role in this structure and cell junction assembly was evaluated using specific inhibitors. The identity of the PARylated proteins was tested by affinity precipitation with PAR-binding reagent followed by western blots. Finally, MCF-7 human breast cancer epithelial cells were subjected to transfection with Tol2-plasmids, carrying a dicistronic expression sequence including Gallus gallus wt VCL (Tol-2-GgVCL), or the same VCL gene with a point mutation in TBM-II (Tol2-GgVCL/*TBM) under the control of a β-actin promoter, plus green fluorescent protein following an internal ribosome entry site (IRES-GFP) to allow the identification of transfected cells without modifying the transfected protein of interest.

Results and discussion

In this work, some of the hypothesis predictions have been tested. We have demonstrated that: (1) VCL TBMs were conserved in vertebrate evolution while absent in C. elegans; (2) TNKS inhibitors disrupted the PAR belt synthesis, while PAR and an endogenous TNKS pool were associated to the plasma membrane; (3) a VCL pool was covalently PARylated; (4) transfection of MCF-7 cells leading to overexpression of Gg-VCL/*TBM induced mesenchymal-like cell shape changes. This last point deserves further investigation, bypassing the limits of our transient transfection and overexpression system. In fact, a 5th testable prediction would be that a single point mutation in VCL TBM-II under endogenous expression control would induce an epithelial to mesenchymal transition (EMT). To check this, a CRISPR/Cas9 substitution approach followed by migration, invasion, gene expression and chemo-resistance assays should be performed.

Introduction

Epithelial morphology, tensile/traction resistance and polarization maintenance rely on intercellular and cell-matrix junctions, where cytoskeleton fibers are anchored. Apart from having structural functions, cell junctions or its molecular components take part in intracellular and intercellular cell signaling nets. An increasing number of nuclear and adhesion complex proteins (NACos) has been recognized. NACos can alternatively be part of the cell junction, or be translocated to the nucleus to work as pro-cell cycling transcription factors, acting as a switch to coordinate cell rounding in mitosis vs. cell anchoring during differentiation (Balda & Matter, 2003; Aho et al., 2009). The adherens junctions epithelial belt—or zonula adherens—allows the anchorage of the subcortical actin ring. Although the complete molecular composition of adherens junctions has been evasive and some authors have suggested the existence of still-not discovered components (Niessen & Gottardi, 2008; Franke, 2009; Carisey & Ballestrem, 2011), E-cadherin, α-catenin and β-catenin have been identified as adherens junctions core proteins. In turn, Vinculin (VCL) links the core proteins to the subcortical actin cytoskeleton. While β-catenin is a well-studied NACo (Balda & Matter, 2003; Aho et al., 2009), nuclear E-cadherin has been detected in lung cancer cells (Su et al., 2015). Interestingly, the adherens junctions epithelial belt is disassembled during a process that facilitates cancer progression which is called epithelial to mesenchymal transition (EMT). EMT involves coordinated changes in cell shape and adhesion, epithelial polarization loss, molecular markers alterations, improved migration and invasion capacity and increased chemoresistance (Kalluri & Weinberg, 2009; Dongre & Weinberg, 2018; Tsubakihara & Moustakas, 2018).

Poly(ADP-ribose) or PAR is a polymer synthesized by poly(ADP-ribose) polymerases (PARPs) from nicotinamide adenine dinucleotide (NAD+), as a posttranslational protein modification. PAR can be lineal or ramified, comprising up to 400 residues. PAR is degraded by poly(ADP-ribose) glycohydrolase (PARG) or other enzymes (Virag & Szabo, 2011; Daniels, Ong & Leung, 2015; Barkauskaite, Jankevicius & Ahel, 2015; Hottiger, 2015). As PAR is rich in phosphates and is negatively charged like nucleic acids, it acts as a “glue” to stabilize protein complexes. Interestingly, specific protein domains act as “readers”, recognizing PAR substructures. The macrodomain identifies terminal ADP-ribose groups, the WWE domain binds iso-ADP-ribose and the PBZ domain recognizes adjacent ADP-ribose groups. In contrast, long and even branched PAR is preferred by RNA recognition motif (RRM) or PAR-binding motif (PBM) (Hottiger, 2015; Leung, 2014). In Vero renal epithelial cells there is a PAR belt associated with the adherens junctions belt that has been evidenced by immunocytofluorescence (ICF) with anti-PAR antibodies. Such PAR belt colocalizes with VCL (Lafon-Hughes et al., 2014). Successive works in the past three decades have implied the VCL pools bound to adherens junctions or focal contacts in the regulation of epithelial cell polarity, adhesion, migration, invasion, and cycling as well as death resistance (Rodríguez Fernández et al., 1993; Pal et al., 2019; Peng et al., 2010; Mierke et al., 2010; Rahman et al., 2016; Bays & Demali, 2017; Rüdiger, 1998; Raz & Geiger, 1982; Coll et al., 1995; Xu, Coll & Adamson, 1998; Sumida et al., 2011; Maddugoda et al., 2007). Recent nuclear VCL detection (Hwang et al., 2017; Flachs, Darasova & Hozak, 2019) indicates that it may behave as a NACo. Cytochalasin D induces the disassembly of the cortical actin ring together with the disruption of the PAR belt. Conversely, if the PARP inhibitor XAV939 is applied at the moment of cell seeding, PAR belt synthesis, cell junction formation and cell adhesion are hampered (Lafon-Hughes et al., 2014). Interestingly, during the EMT, the PAR belt is disassembled together with the adherens junctions epithelial belt (Schacke et al., 2019).

Although the PARP family comprises 18 members in humans (Hottiger et al., 2010; Vyas et al., 2015), only four “true-PARPs” synthesize PAR: PARP-1, PARP-2, Tankyrase-1 and Tankyrase-2 (TNKS-1 & TNKS-2). PARP-1, the canonical family member, is mainly nuclear and has been extensively involved in chromatin remodeling, imprinting and DNA repair (Dantzer & Santoro, 2013; Lafon-Hughes et al., 2008; Vyas et al., 2013), functions that display partial redundancy with those attributed to PARP-2. TNKS can be localized in cell membranes or associated with telomeres and the mitotic spindle proteins (Vyas et al., 2013; Bottone et al., 2012; Chi & Lodish, 2000; Hsiao & Smith, 2009; Bhardwaj et al., 2017). Interestingly, in MDCK renal epithelial cells, overexpressed TNKS-1 is recruited from the cytoplasm to the lateral plasma membrane upon formation of E-cadherin-based cell–cell contacts while it is displaced by extracellular calcium chelation that prevents intercellular adhesion (Yeh et al., 2006), demonstrating its involvement in cell junction dynamics. A role of TNKS in the maintenance of tight junctions and epithelial polarization has been independently demonstrated (Campbell et al., 2016). TNKS recognizes its substrates through the binding of its ankyrin-repeated clusters (ARC) domains to one or more Tankyrase Binding Motifs (TBMs). TNKS binding via TBM is necessary but not sufficient to achieve substrate poly(ADP-ribosylation) or PARylation. Although TBMs were initially described as octamers, the most accepted current view is that a canonical TBM (c-TBM) involves at least a hexamer RXXOXG, where X is any aminoacid and O is Gly, Pro, Ala or Cys. The existence of non-canonical TBMs (nc-TBMs) or indirect interactions with some PARylation substrates was postulated to explain the fact that in certain TNKS KO cells, there is an increase of many proteins which do not contain a c-TBM (presumably due to lack of PARylation/degradation) (Bhardwaj et al., 2017). Non-canonical motifs (nc-TBMs) were later described as heptamers RXXXOXG (Bhardwaj et al., 2017; Guettler et al., 2011; DaRosa, Klevit & Xu, 2018; Pollock et al., 2017; Eisemann, Langelier & Pascal, 2019).

Here, we challenged the hypothesis that TNKS is involved in the synthesis of the PAR belt, with VCL being one of its PARylation targets. More interestingly, we investigated if some VCL functions, like subcortical actin ring anchorage and epithelial cell shape maintenance, depend on its PARylation by TNKS. Sequence analysis disclosed that vertebrate VCL harbors three TBMs, which we identified as c-TBM-I, c-TBM-II (448-453) and nc-TBM-III. The plasma membrane subcellular fraction (PMF) from Vero cells harbored PAR and TNKS and a VCL pool was a PARylation target. TNKS inhibitors disrupted the epithelial belt. Overexpression of VCL with a mutated cTBM-II induced mesenchymal-like cell shape changes in MCF-7 cells. These results provide the first pieces of evidence in favor of our hypothesis and suggest a role for Tankyrase-Binding Motif (TBM) of Vinculin (VCL) in epithelial cells.

Materials and Methods

Cell culture

Vero cells (African green monkey kidney cells, ATCC CCL-81) were routinely cultured in MEM (PAA E15-888 or Capricorn MEM-STA) supplemented with 10% fetal bovine serum (FBS; PAA A15-151 or Capricorn), 100 U/mL penicillin and 100 μg/mL streptomycin (Sigma-Aldrich, St. Louis, MO, USA) at 37 °C and 5% CO2. ATCC MCF-7 cells were received in passage number 6 (P6) as a kind gift from Archana Dhasarathy and Sergei Nechaev, University of North Dakota, USA. MCF-7 cells were cultured up to passage 14 (P14) in Capricorn DMEM-HPSTA with 10% FBS (GIBCO) in an incubator at 37 °C and 5% CO2 or backed up frozen in 5% dimethylsulfoxide (DMSO) and 20% FBS in DMEM-HPSTA.

Protein extraction from subcellular fractions

Subcellular fractions were obtained from control cultures. The plasma membrane protein extraction kit (101Bio, #P503) was used according to the manufacturer’s instructions in order to extract proteins from nuclear (NF), cytoplasmic (CF), internal membrane (IMF) and plasma membrane (PMF) fractions of Vero cells (about 3 × 107 cells). The fractions were then subjected to western blot to detect TNKS and PAR. Fraction controls included Lamin A/C (NF), CALR3 (CF), GRASP (IMF) and PMCA (PMF).

Whole cell protein extraction and affinity precipitation of PARylated proteins

Briefly, cells were grown in a T75 flask, washed, harvested by scraping and resuspended in lysis buffer (50 mM Tris, pH 8, 200 mM NaCl, 1 mM EDTA, 1% Triton X-100, 10% glycerol, 1 mM DTT, 0.1% SDS, and protease inhibitors). The extract was clarified (10 min, 15,000 g, 4 °C), and then subjected to affinity precipitation of PARylated proteins using WWE affinity resin (Tulip #2334) or Af1521 Macrodomain affinity resin (Tulip #2302), according to the manufacturer’s instructions. The resin was equilibrated in lysis buffer (10,000 g, 20 s) and total protein extract was added to the resin pellet and incubated ON in a cold chamber on a rotary shaker. The following day, the resin with the bound PARylated proteins was precipitated by centrifugation and the supernatant was saved as “flowthrough” (FT). The resin pellet was washed three times in lysis buffer. Affinity precipitated proteins (AP) were eluted by incubating in 50 µL protein loading buffer at 99 °C 10 min.

SDS-PAGE and western blot

Protein concentration was estimated by Bradford when appropriate. SDS-PAGE was done as described by Laemmli (1970). Semi-dry protein transfer onto nitrocellulose membranes was achieved using BiometraFastBlot (Analytik Jena). Membranes were blocked with 5% non-fat milk suspension in 0.05% TBS-Tween for 1 h and then incubated with the primary antibody ON and secondary antibody for 1 h. Western blots images were taken using the GeneGnome XRQ Chemiluminescence imaging system after developing with Western Lightning Plus ECL Substrate (Perkin Elmer, Waltham, MA, USA). When necessary, membrane stripping was achieved by incubation in mild Stripping buffer (1.5% Glycine, 0.1% SDS, 0.1% Tween-20, pH 2.2). The antibodies used and their correspondent concentrations are listed in Table 1.

Table 1 Antibodies and counterstains for ICF and WB.

Code	Primary antibody/reagent	ICF	WB	
BD551813	rabbit anti-PAR	1:1,500–1:50	–	
ENZO	rabbit anti-PAR	1:100	1:400	
ENZO BML-SA216	mouse anti-PAR (<50 units)	1:50	1:1,000	
MABE 1031	rabbit anti-PAR reagent	–	1:4,000	
sc-7150	rabbit anti-PARP	–	1:4,000	
GTX117417	rabbit anti-TNKS-1/2 (Nt)	1:200	1:2,000	
Novex 730101	mouse anti-hTNKS; 1106-1125	1:50	–	
ab 73412	rabbit anti-VCL (2016)	1:500	1:1,500	
ab18058	mouse anti-VCL (2013)	1:50	1:1,000	
ab32572	rabbit anti- β-catenin	1:1,000	1:10,000	
ab133597	rabbit anti-E cadherin	–	1:400	
STJ 93885	rabbit anti-lamin A/C	–	1:4,000	
Dr. Gerardo Corradi, IQUIFIB-CONICET	mouse anti-PMCA	–	1:5,000	
STJ190646	rabbit anti-CALR	–	1:2,000	
Dr. Cynthia He Yingxin, National University of Singapore	anti-GRASP	–	1:1,000	
SIGMA	mouse anti-α-tubulin	–	1:10,000	
sc-47778 HRP	mouse anti-actin -HRP	–	1:10,000	
Invitrogen A11122	rabbit anti-GFP	1:1,000	–	
Secondary antibody or counterstain	
ThermoFisher Sci goat anti-mouse 633	goat anti-mouse 633	1:500	–	
Invitrogen A11034	goat anti-rabbit Alexa 488 (A11034)	1:1,000	–	
Invitrogen A11030	goat anti-mouse Alexa Fluor 546 (A11030);	1:1,000	–	
Invitrogen A11029	goat anti-mouse Alexa 488	1:1,000	–	
Invitrogen A11035	goat anti-rabbit Alexa 546	1:1,000	–	
Perkin Elmer	anti-rabbit-HRP	–	1:6,000	
Perkin Elmer	anti-mouse-HRP	–	1:6,000	
ab176756	Cytopainter (actin cytoskeleton)	1:1,000	–	
Sigma–Aldrich	4, 6-diamino-2-phenylindol (DAPI)	1.5 µg/mL	–	

Cell seeding in the presence of Tankyrase inhibitors to monitor epithelial belt assembly.

TNKS inhibitors were kindly provided by Dr. Lari Lehtiö (Biocenter Oulu, Faculty of Biochemistry and Molecular Medicine, University of Oulu, Finland). The reported potencies and IC50 are summarized in Table 2 (Haikarainen et al., 2013; Haikarainen, Krauss & Lehtio, 2014; Riffell, Lord & Ashworth, 2012; Lehtiö, Chi & Krauss, 2013; Mariotti, Pollock & Guettler, 2017; Thorsell et al., 2017; Haikarainen et al., 2016). Vero cells were treated with TNKS inhibitors (TNKSi) at the moment of seeding. Their effect on the PAR belt and on the accompanying cortical actin ring was initially evaluated qualitatively by immunocytofluorescence 5 h after seeding. Later, a quantitative evaluation was carried using the Cell Shape Index (see below). Alternatively, fewer cells were seeded and TNKSi was added on the monolayer and cells were fixed after reaching confluency.

Table 2 Potency of TNKS inhibitors towards PARP-1/2, TNKS-1/2 and TcPARP.

PARP inhibitor	PARP-1	PARP-2	TNKS-1	TNKS-2	
FLALL 9
(compound 8)	>10,000 (Haikarainen et al., 2013)	>10,000 (Haikarainen et al., 2013)	5 (Haikarainen et al., 2013)	>10,000 (Haikarainen et al., 2013)	
MN64	19,000 (Haikarainen, Krauss & Lehtio, 2014)	35,000 (Haikarainen, Krauss & Lehtio, 2014)	6 (Haikarainen, Krauss & Lehtio, 2014)	72 (Haikarainen, Krauss & Lehtio, 2014)	
XAV 939	2,200 (Riffell, Lord & Ashworth, 2012; Lehtiö, Chi & Krauss, 2013)
5,500 (Mariotti, Pollock & Guettler, 2017)
74 (FL) (Thorsell et al., 2017)	27 (Thorsell et al., 2017) (FL)	11–13 (Riffell, Lord & Ashworth, 2012; Lehtiö, Chi & Krauss, 2013)
5 (Mariotti, Pollock & Guettler, 2017)
95 (Thorsell et al., 2017)	4–5 (Riffell, Lord & Ashworth, 2012; Lehtiö, Chi & Krauss, 2013)
2 (Mariotti, Pollock & Guettler, 2017)
5 (Thorsell et al., 2017)	
G007LK	–	–	46 (Haikarainen et al., 2013)	25 (Haikarainen et al., 2013)	
OD35
(compound 2)	–	–	11,000 (Haikarainen et al., 2016)	260 (Haikarainen et al., 2016)	
Note:

In vitro IC50 values determined for activity assays using full length (FL) or truncated recombinant enzymes. All concentrations are expressed as nanomolar (nM).

Indirect immunocytofluorescence

Indirect immunocytofluorescence was carried as in Lafon-Hughes et al. (2014) on 12 mm round cover glasses placed in 24-well plates. Briefly, cells were fixed in 4% paraformaldehyde (PFA), permeabilized and subjected to blocking. Then, they were incubated with specific primary antibodies (see Table 1) diluted in blocking buffer (0.2% Tween, 1 % BSA in fPBS) for 2 h at 37 °C. After washing, cells were incubated (1 h, RT) with the correspondent anti-antibody mix in blocking buffer and/or cytopainter (ab176756). After washing, nuclei were counterstained with DAPI, coverslips were mounted in Vectashield (Vector 94010) or Prolong Gold (Molecular Probes P36930) and sealed. In order to check the specificity of the signals, controls without primary antibodies were carried in parallel.

Epifluorescence and confocal microscopy

Images were recorded with an epifluorescence microscope (40× objective) with DP controller software, an Olympus BX61/FV300 (Tokyo, Japan) with a Plan Apo 60 x/1.42 NA oil immersion objective or a ZEISS LSM800-Airyscan (Oberkochen, Germany) confocal microscope. Original images were taken in the same conditions as reference images of controls without primary antibodies at the same microscopy session. All images in each experimental series were taken with the same setting at the same confocal session. If modified, all were subjected to the same degree of brightness/contrast adjustment and Gaussian blur filtering, including the control without a primary antibody. The ImageJ free software was used for image processing.

VCL vs. PAR colocalization in epithelial belts: overlapping pixels highlight and quantification examples

Colocalization can be analyzed in terms of signals overlap or signals intensity correlation.

Overlapping pixels highlight

Colocalization was initially evidenced qualitatively by visual inspection. As PAR signal was usually lower than VCL signal, the signal overlap visualization was facilitated through the creation of masks. Masks creation involves the selection of an intensity threshold above which there is a real signal and below which there is noise. The same threshold was used in controls without primary antibody (obtaining empty masks) as an extreme caution to avoid taking noise as signal. As masks are segmented images relying on values of 1 or 0 in each pixel, the multiplication of masks images (using Image J/Image Calculator) allowed a perfect outline of the overlapped regions (in yellow).

Exemplar overlapping extent quantification

Mander’s indexes M1 and M2 calculated using masks reflected the overlapping extent: the proportion of positive pixels in channel A that also had a positive signal in channel B or vice versa (the proportion of positive pixels in channel B that harbored a positive signal in channel A).

Exemplar intensity correlation analysis

Pearson Coefficient (Rr), which ranges from −1 to +1 (from anti-colocalization to complete colocalization) and Li’s ICQ (which ranges from −0.5 to +0.5) were calculated on ROIs (regions of interest). PDMs (products of the differences from the means) were localized. PDMs can be double positives (both above the mean), represented as ++PDMs, or double negatives (both below the mean). ++PDMs were mapped on the image (in violet).

Quantitative measures by FRET on indirect ICF images (see Article S1)

Relative expression of VCL in cells transfected with Tol2-GgVCL vs. Tol2-VCL/*TBM

Intensity quantification on confocal images was used to estimate the relative expression of VCL and GFP in transfected cells. Whole-cell contours and nuclear contours were used as ROIs to determine whole-cell GFP, whole-cell VCL, nuclear GFP and cytoplasmic VCL.

Cell shape index: combined roundness and circularity

Epithelial cells display tension in their adherens junctions belts and subcortical actin rings, giving rise to straight lines. Besides, even in the absence of neighbor cells, they have quite sharp borders. In contrast, mesenchymal or mesenchymal-like cells have serrated, wavy junctions and an irregular outline due to the presence of filopodia and lamellipodia (which can be decorated with ruffles) (Innocenti, 2018). To be able to quantify and compare morphological changes, it was required to express these morphologies in terms of numbers. ImageJ offers measurements of Roundness and Circularity. While Roundness accounts for gross changes (from circular to ovoid or triangular), it is insensitive to straight vs. serrated or undulated borders. On the other hand, Circularity is very sensitive to such edge characteristics. Thus, we defined a Cell Shape Index (CShin) combining Roundness and Circularity.

CShin=RoundnessCircularity

Figure 1A depicts values of CShin associated to some geometric figures (on the left). CShin is sensitive to radial symmetry, being higher for a circle than for an ellipse or higher for a square than a rectangle. If the clean edge of each geometric form is substituted by a serrated or undulated border, CShin values show a sharp increase. As can be seen in Fig. 1A (right), CShin is very sensitive to the edge characteristics of the shape. For example, CShin = 1 for a perfect circle and CShin = 1.81 for a form that conserves the circular proportions but has an undulated edge.

Figure 1 Cell shape index: design and proof of concept.

(A) Roundness, circularity and their quotients were initially measured and calculated for geometric figures. Here are Cshin values for some geometric figures and the correspondent figures with wavy limits. (B) NMuMG cells are a well-known model to study a process through which an epithelial cell loses its epithelial characteristics and acquires mesenchymal characteristics. Such process is an epithelial to mesenchymal transition (EMT) and is induced by treatment with TGF-β. We induced that process and measured Cshin to validate the proposed index. ****p < 0.0001.

As a proof of concept, CShin was applied to a well-characterized cell model of an epithelial to mesenchymal transition (EMT). NMuMG (NAMRU Murine Mammary Gland) cells were measured untreated (when they have epithelial characteristics) or after treatment with TGF-β (when they have mesenchymal characteristics).

The Plotly Chart Studio online graph maker (https://chart-studio.plotly.com/create/box-plot/#/) was used to build box & whisker graphs, plus represent all the individual data points. Statistics were done using the free software available online at https://astatsa.com/, attributed to Vasavada (2016). As CShin statistical distribution is not normal, medians were compared using Mann-Whitney non-parametric test. Kruskall-Wallis non-parametric ANOVA followed by Conover test adjusted by Benjamini-Hochberg FDR method.

The shape change was evidenced as an increased CShin index (Fig. 1B). In this model, CShin mean and median were below 1 for epithelial cells and above 1 for mesenchymal cells, with a significant difference (p = 2.413887 × 10−8). Thus, the index proved to be useful to evidence the cell shape change from an epithelial to a mesenchymal cell.

TBM analysis in hVCL and homologs

Using SnapGene viewer, hVCL protein RefSeq sequence was searched for the c-TBM, RXXOXG, and the nc-TBM, RXXXOXG, where X is any aminoacid and O is Gly, Pro, Ala or Cys. After global lineal-sequence alignment was made using ClustalX 2.0 for Windows (Larkin et al., 2007), TBM conservation was checked among VCL homologs in different representative species. The Reference Sequences used were (as in Han, Van der Krogt & De Rooij, 2017): Human NP_003364.1, Mouse NP_033528.3, Chicken NP_990772.1, Xenopus NP_001090722.1, Zebrafish NP_001122153.1, Drosophila NP-476820.1 and C. elegans NP_501104.2). Comparison between hVCL and dVCL sequences were made using ExPASy alignment tool, SIM (Huang & Miller, 1991). Result indicated as Drosophila 1 corresponds to the comparison matrix PAM400, while result indicated as Drosophila 2 corresponds to the comparison matrix BLOSUM30, using equal parameters in both cases.

Drosophila VCL structure was predicted using CPHmodels-3.0 (Nielsen et al., 2010). Structural alignment and Figures were made using YASARA (Krieger, Koraimann & Vriend, 2002).

Generation of eukaryotic expression vectors with WT-GgVCL (Tol2-GgVCL/WT) or G454V-GgVCL (Tol2-GgVCL/*TBM)

To disrupt the association of VCL with TNKS, Gly454 (in TBM-II) was substituted for a Val residue. A ggc codon was substituted for gta by directed/circular mutagenesis using custom primers G454V (Table 3) and following the protocol published by Wang and Wilkinson (Wang & Wilkinson, 2000). pET15b/GgVcl 1-1066 (Addgene #46171) bearing the complete Gallus gallus VCL sequence was used as template. An AccI (BioLabs R0161S) (gtagac) site generated by the point mutations introduced was used as a diagnostic recognition site. The new plasmid, pET15/GgVcl/*TBM (Addgene #162781) is available at https://addgene.org/162781/.

Table 3 Primer sequences.

Mutagenic primers
G454V (#10336022: P180917-006 A01 & A02)	Fwd:CTGCGACGACATGGGAAAGTAGACTCTCCTGAGGCCCGT	
Rev:ACGGGCCTCAGGAGAGTCTACTTTCCCATGTCGTCGCAG	
Primers with Kozak and restriction enzymes cut sites to facilitate transference to the eukaryotic plasmid
ch-VCL (#10336022:379546 A01 & A02)	Fwd (KpnI-Kozak-chVCL):
5->3 : ATATggtaccACCATGatgcccgtcttc (28 nt)	
RV (EcoRI-STOP-ch VCL):
5->3: AGCTgaattcctaCTGATACCATGGGGTC (29 nt)	

The GgVCL and GgVCL/*TBM ORFs were then cloned into TOPO TA #450640 through the topo cloning reaction, following the manufacturer’s instructions, and then into a middle entry plasmid with multiple cloning sites (Tol2 Kit#237, pME-MCS), to obtain the pME-VCL vector.

The final mammalian expression vector was obtained through the use of Tol2 kit LR (Kwan et al., 2007; Yagita et al., 2010). The LR reaction involved the use of Gateway LR Clonase II Plus Enzyme Mix (Invitrogen 12538-120) to fuse 4 plasmids into a single final vector: the 5′entry plasmid with a β-actin promoter called p5E-actin (#299), the middle entry plasmid with the VCL insert pME-VCL (wt or mut), the 3′entry plasmid p3E-IRES-nls-GFP (#391) and the destination vector pDestTol2pA2 (#394). The resulting 14.5 KB eukaryotic expression vectors carried the WT or mutant VCL under a β-actin promoter and a GFP coding sequence driven by an internal ribosome entry site (IRES-GFP). Tol-2 GgVCL and Tol-2 GgVCL/*TBM constructs were amplified in NEB C3019I bacteria and corroborated by primer walking sequencing (Macrogen, Seoul, South Korea). Both Tol-2 plasmids (Addgene #162787 and #162790) are available at https://addgene.org/162787/ and https://addgene.org/162790/.

Transfection of MCF- 7 cells with Tol2-GgVCL or Tol2-GgVCL/*TBM

FUGENE transfection reaction: DNA (3:1) was used to transfect MCF-7 undisrupted monolayers. Cells were monitored under epifluorescence and phase contrast or subjected to ICF at the indicated times. 1 μg Pmax GFP plasmid (~3 KB) from Amaxa Mouse/Rat Hepatocyte Nucleofector kit was used as a positive control to evaluate transfection efficiency of a DNA plasmid.

Results

VCL sequence harbors three TBMs conserved in vertebrates

All confirmed TNKS substrates to date contain one or more TBMs that are recognized via TNKS ARC domains (ARC1, ARC2, ARC4 and ARC5) (DaRosa, Klevit & Xu, 2018). Therefore, proteins harboring a TBM are putative TNKS PARylation targets. We examined the presence of TBMs in hVCL. Also, a comparative alignment was done with VCL sequences from model organisms including some vertebrates and two invertebrates: Drosophila melanogaster, whose adherens junctions are quite similar to those of vertebrates, and Caenorhabditis elegans, whose adherens junction proteins are not essential for general cell adhesion or for epithelial cell polarization.

Human VCL has three TBMs (Fig. 2A): TBM–I is RARGQG at positions 339-344 (exons 8-9), TBM–II is RRQGKG at 449-454 (exons 10-11) and TBM–III is RGLVAEG at positions 520-526 (exon 12). The first two sites are canonical (RXXOXG) and correspond to peripheral, easily accessible loops whereas the third is non-canonical (RXXXOXG) and is situated in a deeper position, on a helix.

Figure 2 VCL sequence harbors three TBMs conserved in vertebrates, one of which is present in Drosophila.

(A) TBMs, according to the hexamer rule for c-TBMs or the heptamer rule for nc-TBMs, follow the patterns: R XXOX G or R XXXOX G, where X is any aminoacid and O is G, P, A or C. These two patterns were searched and found conserved in vertebrates (Human, Mouse, Chicken, Xenopus and Zebrafish) as indicated under TBM–I, TBM–II and TBM–III. Only one of them was present in Drosophila while none of them was present in Caenorhabditis elegans. The single TBM present in Drosophila can align with human TBM–I or TBM–II depending on the matrix used for the alignment. (B) Superimposition of hVCL (blue, PDB:1TR2, B subunit) and dVCL predicted structure (grey). TBMs from hVCL are highlighted using the same color code as in A, while Drosophila TBM is colored in pink. Overlapped hVCL and Drosophila VCL structures evidence the equivalence of Drosophila TBM and vertebrate TBM–II.

All TBMs were conserved in other vertebrates such as Mus musculus, Gallus gallus, Xenopus laevis and zebrafish, while only one of them was also found in D. melanogaster. In contrast, none of them was present in C. elegans (Fig. 2A). It is interesting to note that the single TBM domain of Drosophila could align with both canonical TBMs (TBM I and II) of hVCL. Hereafter, structural comparison between hVCL and dVCL was done. The superimposition of hVCL with the predicted 3D-structure of dVCL indicated that Drosophila TBM was located in the same relative position in the structure as the human VCL TBM–II (Fig. 2B).

Endogenous PAR and TNKS were detected in the plasma membrane fraction of Vero cells

PAR had been previously detected associated to the epithelial belt in Vero cells by ICF using an anti-PAR antibody (Lafon-Hughes et al., 2014). Now we corroborated PAR presence in subcellular fractions whose purity was checked with specific markers (Fig. 3A). Instead of an antibody, the novel anti-PAR reagent MABE 1031 was used to evidence the presence of PAR associated to the plasma membrane fraction (PMF). MABE 1031 (AB_2665467) is a recombinant protein comprising the WWE domain from RNF146 and the Fc region of rabbit IgG. The latter is recognized by anti-rabbit secondary antibodies. Thus, it can be handled as an equivalent to a primary antibody in ICF, WB or affinity pull-down applications.

Figure 3 Endogenous TNKS and PAR distribution in epithelial cells.

(A) Confluent Vero cells were subjected to subcellular fractionation. The identity of the obtained fractions was confirmed using fractionation validation markers: for nuclear fraction (NF), Lamin A/C; for cytoplasmic fraction (CF), CALR3; for internal membrane fraction (IMF), GRASP; and for plasma membrane fraction (PMF), PMCA. TNKS was more abundant in IMF, but was also detected in the other cell fractions, including the PMF. PAR (detected with anti-PAR reagent MABE 1031) was more abundant in the nuclear fraction but also detectable in the other cell fractions. Arrows on the right indicate discrete bands corresponding to unknown PARylated proteins observed in the PMF. (B–E) ICF to detect TNKS cellular localization (green), counterstained with F-actin (red) and DAPI (blue). A non-symmetric perinuclear region probably correspondent to the Golgi system harbored most TNKS. However, (D, E) a closer examination evidenced a wider TNKS distribution, including plasma membrane or submembrane regions, as indicated by white arrows. Bar: 25 µm.

TNKS presence in the internal membrane fraction (IMF) was conspicuous and a slighter signal was also detected in nuclear fraction (NF), cytoplasmic fraction (CF) and plasma membrane fraction (PMF) (Fig. 3A). The cell fractionation results matched with ICF images (Figs. 3B–3E). In ICF, a TNKS ‘hat’ could be observed on one side of the nucleus, presumably corresponding to the Golgi complex. It was possible to see that a mild/slight TNKS signal spread towards the rest of the cell, reaching the plasma membrane or subcortical region sometimes (Figs. 3D, 3E). This TNKS distribution was robust, since it was observed in ICF experiments performed using two different anti-TNKS antibodies designed to recognize the Nt or the Ct (catalytic domain), as well as under two different fixation protocols (Fig. S1).

TNKS inhibitors disrupted the PAR belt synthesis and altered cell shape

A tight adherens junctions belt, indicative of tension, involves a straight PAR belt and a straight subcortical actin belt. In contrast, an incomplete lax belt is associated with punctuated (less abundant) or serrated PAR and F-actin signals.

Four TNKS inhibitors with different potency and specificity towards TNKS-1 and TNKS-2 (Table 2), were added to cell cultures at the moment of seeding (Fig. 4). Results were ordered according to TNKSi in vitro specificity, starting with TNKS-1-prone FLALL (Figs. 4B, 4G, 4L), then MN64, G007LK and the TNKS-2 selective OD35 in the last position (Figs. 4E, 4J, 4O). All of them hindered PAR belt synthesis. A punctuated or zipper-like discontinuous distribution of PAR was observed (Figs. 4B–4O). The PAR belt did not completely disappear, which could be explained by incomplete TNKS inhibition. Cortical F-actin changed accordingly (Figs. 4G–4J), reinforcing our previous results correlating PAR and F-actin distributions in epithelial cells and nerves (Lafon-Hughes et al., 2014; Lafon Hughes et al., 2017). In the presence of MN64, less cells remained attached after the PBS wash done previous to fixation. Such a cell density decrease has been previously documented in the presence of a TNKS and PARP-1/2/3 inhibitor called XAV939 (Lafon-Hughes et al., 2014).

Figure 4 TNKS inhibitors hampered PAR belt synthesis and altered the subcortical actin ring.

TNKSi were added to Vero cells at the moment of seeding and after 5 h cells were fixed, washed in PBS and subjected to ICF to detect PAR (green), F-actin with rhodamine-phalloidin counterstain (red) and nuclei with DAPI (blue). (A–E) PAR channel; (F–J) F-actin channel, (K–O) merged channels. (A, F, K) control, (B, G, L) 80 nM FLALL-9, (C, H, M) 60 nM MN-64, (D, I, N) 250 nM G007-LK, (E, J, O) 2.6 μM OD35. Instead of the tight straight continuous appearance of the belt (A, F, K), a serrated discontinuous distribution was observed (B–O). Bar: 25 µm. For a quantitative expression of cell shape change, see Fig. 5.

Figure 5 TNKSi changed cell shape.

Cells were exposed to the PARP-1/2-3 inhibitor Olaparib (OLA) or to TNKSi FLALL, MN64, G007LK or OD35, at the moment of seeding. Five hours later, they were fixed after a quick wash and subjected to ICF (see images in Fig. 4). Cell shape index was calculated as: CShin = Roundness/Circularity. To appreciate the overall behavior of CShin and its validation as an index, see Fig. 1. ***p < 0.001; ****p < 0.0001.

With the aim of quantifying changes in cell shape, particularly those involving cell contours, we designed a Cell Shape Index or CShin (see “Materials and Methods”, section 9, Fig. 1).

Median CShin (Fig. 5) was below 1 in control cells, as expected for epithelial cells according to the index validation in NMuMG cells (Fig. 1B). No effect of the PARP-1/2/3 inhibitor Olaparib (OLA) was detected. In contrast, all TNKSi, namely FLALL, MN64, G007LK and OD35, induced a significative increase of CShin, indicating a measurable change in cell contours.

We did not find a correlation between TNKSi selectivity and the observed effects on PAR belt, the subcortical F-actin cytoskeleton and CShin. The potency and specificity of the inhibitors were determined in vitro while, in whole cells, factors as permeability, subcellular distribution, stability and metabolic alterations may affect the effective dose and outcome. Therefore, a straight-forward correlation was not necessarily expected.

As stated in the introduction, in epithelial cells the subcortical actin ring and the PAR belt are somehow associated. We are now arguing that PARylation of a target protein joining adherens junctions and the F-actin cytoskeleton by TNKS1/2 is at least one of the factors that could explain this association.

A VCL pool was a PARylation target

Using a WWE-affinity resin, covalently PARylated proteins were affinity precipitated. Next, cell junction or cytoskeletal proteins were tested by WB as putative PARylation targets (Fig. 6A). Interestingly, while β-catenin, E-cadherin, actin and α-tubulin were recovered only in the flow through (FT), a fraction of VCL was affinity precipitated (AP) by the WWE-affinity resin which consists of highly purified GST-RNF146(100-175) fusion protein bound to glutathione beads. Only anti- α-tubulin and anti-VCL antibodies were of mouse origin; thus, the result is clear-cut and cannot have been affected by successive membrane stripping, demonstrating the existence of a PARylated VCL pool. Given the AP was done in the absence of crosslinking and in the presence of high detergent and salt concentrations, non-covalent interactions were not expected to be maintained. This was reflected in the fact that other proteins of the adherens junctions complex were not affinity precipitated. Thus, this result indicated the existence of covalently PARylated VCL.

Figure 6 A VCL pool was PARylated.

(A) Vero cell proteins were affinity precipitated using PAR-recognizing WWE-resin to recover covalently PARylated proteins. The flowthrough (FT) and affinity precipitated fractions (AP) were subjected to WB. Five cell junction and cytoskeletal proteins were assayed. Unlike β-catenin, E-cadherin, actin and α-tubulin, a VCL pool was affinity precipitated (the arrow points at the lower apparent MW VCL band, around 90 to 100 KDa, which was the most abundant in the AP fraction). As expected, the resin did also precipitate PARylated PARP-1 and PAR itself. The species in which each antibody was raised is indicated under the correspondent label (R: rabbit; M: mouse). Except for mouse anti-α-tubulin and mouse anti-VCL, all primary antibodies used (anti-β-catenin, anti-E-cadherin, anti-PARP-1, anti-PAR) were of rabbit origin or directly labeled (anti-actin). (B) Diagram indicative of the relative position of confocal images of the right panels (C–H): ICF anti-VCL (red) and anti-PAR (green), merged channels; (I–N) correspondent VCL and PAR masks; (O–T): in yellow, masks product highlighting the overlapping VCL and PAR regions, drawing precisely the epithelial belt. Subsequent confocal planes, from basal to apical, allowed the distinction of VCL in focal adhesions (e.g., C, I, O) from VCL associated to the PAR belt (e.g., G, M, S). Bar: 25 µm. (U, V): VCL and PAR region of interest (ROI) around the belt (cropped from E). (W): ++PDMs from an intensity correlation colocalization analysis. The pixels with values above the mean of each channel, in both channels, are depicted. (X): in yellow, the product of U and V masks (as in (O–T)). Notice that the complex intensity correlation analysis and masks images multiplication lead to the same result: colocalizing PAR and VCL pixels are on the epithelial belt. (Y): enlarged belt masks ROIs extracted from (L–N), facilitate the precise visualization of colocalizing pixels. Overlapping percentages were calculated according to Mander’s coefficients (small numbers or results section).

VCL was also affinity precipitated as a light band in experiments carried using the Af1521 Macrodomain resin, which recognizes mono or poly(ADP-ribosylated) proteins. PARP-1 was also checked as an AP control. While the specific TNKSi G007LK hindered only VCL affinity precipitation (Fig. S2A), XAV939, which inhibits PARP-1/2/3 & TNKS, hampered both VCL and PARP-1 affinity precipitation (Fig. S2B).

Altogether, these results support that TNKS is responsible for VCL PARylation.

To map the subcellular localization of PARylated VCL, we evaluated the co-occurrence of PAR and VCL signals in confocal ICF images. Figure 6B is a scheme representing the relative position of the subsequent confocal epithelial monolayer sections, running from basal to apical direction shown as a merged channels montage in Figs. 6C–6H. The correspondent masks and multiplied masks montages (Figs. 6I–6N and Figs. 6O–6T) are also displayed to highlight the overlapping points. VCL was present in all these sections but the PAR belt was distributed along a 1 to 1.5 µm in z (3 to 5 slices depending on voxel height). PAR and VCL signals overlap drew just a thin polygon (in this case, a pentagon) representing the epithelial belt. No PARylation was ever detected at VCL-rich focal contacts (neither in untreated or treated cell cultures of Vero, NMuMG or MCF-7 cells, nor in mice epithelial tissues).

To quantify these observations, colocalization was analyzed in terms of signals overlap or signal intensity correlation (see “Materials and Methods”).

We have included an example of intensity correlation analysis, taking a ROI from Fig. 6E. The ROI red and green channel images (Figs. 6U and 6V) were subjected to an intensity correlation analysis. Pearson Coefficient (Rr), which ranges from −1 to +1 (from anti-colocalization to complete colocalization), was 0.191. Li’s ICQ (which ranges from −0.5 to +0.5) was 0.235. Double positive PDMs (products of the differences from the means) were localized precisely in the belt region (Fig. 6W), just as the mask obtained through the correspondent masks multiplication (Fig. 6X). Therefore, the intensity correlation analysis and the overlapping analysis converged in the epithelial belt region. Thus, mask multiplication to highlight overlapping signals and the overlapping analysis was used hereon (see Figs. S3A–3SL). We have additionally explored PAR and VCL colocalization in an EMT model. Epithelial NMuMG cells display PAR and VCL colocalization (overlap) precisely at the epithelial belt region (Fig. S3M) which is lost upon treatment with TGF-β to induce EMT (Fig. S3N) (for further reference, see our previous work (Schacke et al., 2019)). We have also analyzed PAR-VCL in belt ROIs colocalization (overlap) focused on epithelial belt ROIs (Fig. 6Y). In this case, Mander’s coefficient M1 ranged from 0.262 to 0.660, indicating that ≈ 30% to 60% of belt VCL signal overlapped with PAR signal; M2 ranged from 0.255 to 0.882 indicating that ≈ 30% to 90% of belt PAR signal overlapped with VCL.

Partial colocalization of PAR and VCL (within 200 nm) had also been previously observed in Vero cells by ICF and confocal microscopy (Lafon-Hughes et al., 2014). FRET was carried after indirect ICF (Fig. S4), lowering the colocalization maximal distance to about 50 nm (estimated according to König et al 2006 (Konig et al., 2006)) as the addition of 10 nm FRET + 10 nm each primary antibody + 10 nm each secondary antibody. FRET results were not included in the main manuscript because this resolution did still not allow to distinguish among VCL and nearby PARylated proteins nor among covalently PARylated or PAR-bound proteins; thus, the affinity precipitation result constituted by far stronger evidence.

Not only not all VCL was PARylated (according to AP and ICF results), but also not all belt PARylation could be attributed to VCL; the band pattern detected after subcellular fractionation using the PAR-detecting reagent indicated there may be additional targets.

Mesenchymal-like shaped cells were observed under VCL/*TBM transient overexpression

We finally wanted to study whether VCL PARylation was related to its functions. Using the obtained information and given the importance of the single conserved c-TBM in Drosophila to preserve its function, we decided to mutate the equivalent motif (c-TBM-II) in vertebrate VCL.

Direct mutagenesis of VCL in pET15b/GgVCL was achieved using primer pair G454V and the circular mutagenesis protocol suggested by Wang and Wilkinson (2000) (Wang & Wilkinson, 2000) in the presence of DMSO. As depicted in Fig. 7A, a ggc codon (coding Glycine) was substituted by gTA (coding Valine), generating an AccI (BioLabs R0161S) diagnostic restriction recognition site (gTAgac). Then, after incorporating each VCL sequence to a middle entry plasmid (pME), pME-VCL and pME-VCL/*TBM were obtained. The recombination of a plasmid carrying a β-actin promoter, pME-VCL or pME-VCL/*TBM, a plasmid coding an IRES-GFP and a destination plasmid allowed to obtain the desired eukaryotic expression plasmids depicted in Fig. 7B.

Figure 7 Transfection of MCF-7 (P10) monolayer with Tol2 14.5 KB eukaryotic expression vectors harboring VCL or VCL/*TBM.

(A) Circular PCR was carried on the 8909 pb plasmid template pET15b/GgVCL using G454V mutagenic primers to generate pET15b/GgVCL/*TBM (B) Schematic representation of the 14.5 KB eukaryotic expression plasmids Tol2/Gg-VCL and Tol2/Gg-VCL/*TBM, expressing the correspondent WT or mutated VCL under a β-actin promoter, plus green fluorescent protein whose expression relied on an Internal Ribosome Entry Site (IRES-GFP). In this way, transfected cells were expected to harbor free GFP (being easily identifiable) while expressing untagged VCL.

Transfection of MCF-7 cells with Tol2-GgVCL (Figs. 8A, 8B, 8E–8J) or Tol2-GgVCL/*TBM (Figs. 8C, 8D, 8K–8O) was achieved using FUGENE reagent on cell monolayers. These Tol2 plasmids allow the identification of transfected cells that express the plasmid as GFP+ cells (Figs. 8A′–8O′, in green). Although carried in parallel and in the same conditions, the transfection with Tol2-GgVCL/*TBM rendered more GFP+ cells than the WT version, indicating that this transfection was more effective or less cytotoxic (low cell numbers did not allow carrying additional quantifications to distinguish among these two scenarios). Moreover, image intensity quantifications showed that GFP expression was similar in Tol2-GgVCL or Tol2-GgVCL/*TBM cells while VCL expression was doubled in the mutant with respect to the WT transfected cells. It is worth noting that cells transfected with Tol2-GgVCL/*TBM presented a distorted morphology without lineal tense cell junctions and with filopodia or lamellipodia, that looked mesenchymal-like (Figs. 8C, 8D, 8K–8O). Cell shape was measured and the results are in Fig. 9. While GgVCL transfected cells displayed a median CShin <1, cells transfected with GgVCL/*TBM had a significatively higher CShin. This result strongly suggested the involvement of TBM in the function of VCL to maintain cell shape.

Figure 8 Transfection with Tol2-GgVCL or Tol2-GgVCL/*TBM.

MCF-7 (P10) monolayers were transfected with either Tol2-GgVCL or Tol2-GgVCL/*TBM. Cells were fixed 17 h later. GFP (green) and VCL (red) were detected by ICF. The left half of the figure (A, B, E–J) corresponds to Tol2-GgVCL (WT); the right half (C, D, K–O) corresponds to Tol2-GgVCL/*TBM. (A–O): mixed channels; (A′–O′): only GFP; (A″–O″): only VCL. Notice the abundant VCL expression (neighbor cells in A are hardly detected) and the irregular, mesenchymal-like shape of cells harboring mutated VCL. Bar: 25 µm.

Figure 9 Cell shape was different in cells transfected with WT or TBM*-GgVCL.

Cell shape index after transfection with WT VCL was below 1, as in other epithelial cells, while transfection with VCL/*TBM significantly increased the index value. **p < 0.01.

To confirm this result, we aimed to use a KO-add back approach. We attempted to obtain VCL knocked-out MCF cells by a CRISPR/Cas9 strategy (Fig. S5A). Only a limited heterogeneous cell pool could be obtained as transfected cells grew extremely slowly and lost their colony forming ability under isolation (Figs. S5B–S5Q, Figs. S6A–S6H), precluding the obtainment of pure clones and the possibility of straight-forward VCL KO confirmation. The available cells, however, were transfected with the Tol2-GgVCL/*TBM plasmids (Figs. S7A–S7N) or Tol2-GgVCL (Figs. S7O–S7R). CShin was sensitive enough to detect changes in cell shape induced by the presumptive VCL partial “KO” which were reverted by the transfection with WT GgVCL but not with Gg VCL/*TBM (see contours examples in Figs. S7E, S7I, S7L & S7P, and the graph in Fig. S8). Unfortunately, the obstacles regarding the “KO” cells obtainment limited the conclusions that could be driven from these experiments and pointed out the need to implement an alternative approach to confirm these observations.

Discussion

Our hypothesis was that TNKS PARylates epithelial belt VCL, affecting adherens junctions VCL-dependent functions, like subcortical actin ring anchorage and epithelial cell shape maintenance. The insight to consider this pool of VCL as a putative PARylation target came from ICF images and data obtained in Vero cells, where partial colocalization of belt PAR and VCL was observed (Lafon-Hughes et al., 2014), indicating a distance <200 nm among them. FRET allowed to shorten the spatial proximity radio from PAR to VCL (or from VCL to PAR) to <50 nm (Fig. S4). The hypothesis that TNKS could be the PARylating enzyme was based on previous works indicating the role of TNKS in cell junctions (Yeh et al., 2006; Campbell et al., 2016). We performed a sequence analysis that revealed that human VCL harbored three TBMs: two c-TBM and one nc-TBM. In 2011 Guetler and collaborators reported a huge list of TMB bearing proteins. VCL was included, but at that time only the canonical TBMs were reported (Guettler et al., 2011). TBMs presence is a requirement to be a TNKS target. For this reason, the presence of TBMs reinforced the hypothesis regarding both VCL as a PARylation target and TNKS as the involved enzyme. Moreover, the three detected TBMs were conserved among well-studied vertebrate models including green monkey, mice, chicken and zebrafish. Among invertebrates, one of the TBMs was present in VCL from Drosophila, whose adherens junctions are quite similar to those of vertebrates, while none of the TBMs was found in Caenorhabditis elegans. Interestingly, “in contrast to vertebrates, C. elegans adherens junction proteins are not essential for general cell adhesion or for epithelial cell polarization” (Armenti & Nance, 2012). Thus, a comparative analysis suggested that, although the presence of more than one TBM may facilitate the establishment of stronger interactions with TNKS, at least one of the VCL TBMs was deeply involved in adherens junctions functions and was highly conserved. A simple sequence alignment of human/vertebrate vs. Drosophila VCL yielded two possible results, pointing to TBM-I or TBM-II (Fig. 2A). In silico modeling of Drosophila VCL allowed us to observe that TBM-II in human VCL is a loop localized like Drosophila single TBM (Fig. 2B).

The binding of a substrate protein by ARCs is required for subsequent PARylation, but may not be sufficient for the modification, as certain substrates are not modified upon binding.

PAR association with the plasma membrane was confirmed through a biochemical approach, using the PAR-binding reagent MABE 1031 to detect PAR in a plasma membrane fraction (PMF) of Vero cells (Fig. 3A). These results agree with the recent detection of enriched cortical PAR during mouse oocyte meiosis (Xie et al., 2018) and confirm the ones previously obtained combining ICF with either post-fixation PAR digestion with PARG or pretreatments of cells with XAV939 in Vero cells (Lafon-Hughes et al., 2014). XAV939 was formerly thought to be a specific TNKS inhibitor (Riffell, Lord & Ashworth, 2012; Lehtiö, Chi & Krauss, 2013; Haikarainen, Krauss & Lehtio, 2014). Therefore, data relying on XAV939 were interpreted as favoring a role of TNKS-1/2 in PAR belt synthesis (Lafon-Hughes et al., 2014) that fitted with the results obtained overexpressing TNKS in MDCK cells (Yeh et al., 2006). Nevertheless, it has been recently demonstrated that XAV939 does also inhibit full length PARP-1 and PARP-2 (Thorsell et al., 2017), raising again the question of which is the PARP responsible for the synthesis of the PAR belt. Experiments with more specific TNKS-1/2 inhibitors supported TNKS involvement in PAR belt synthesis (Fig. 4). Additionally, TNKS was detected in the plasma membrane fraction of Vero cells (Figs. 3A–3D).

Finally, affinity precipitation of covalently PARylated proteins was done using WWE affinity resin (Fig. 6A) or Macrodomain resin in the absence and presence of TNKSi (Figs. S2A, S2B). Clear-cut results were obtained demonstrating that a VCL fraction was precipitated while cortical actin or other components of adherens junctions, like E-cadherin or β-catenin, were not. Both the experimental conditions (precipitation under extremely astringent conditions and in the absence of cross-linking) and the fact that VCL partners from adherens junction complexes were not affinity precipitated argue in favor of a covalent VCL PARylation. This, together with the fact that VCL harbors TBMs and that cell shape changes if a single aminoacid in a single TBM is substituted, precluding a putative PARylation by TNKS, demonstrates that VCL is a direct TNKS target. VCL had formerly been detected in PAR-bound protein pools. Gagné et al (Gagné et al., 2008; Gagné et al., 2012) detected hVCL (but not E-cadherin or β-catenin) in fractions immunoprecipitated with anti-PAR antibodies followed by mass spectrometry (Gagné et al., 2008; Gagné et al., 2012). Moreover, VCL was recently identified in a pool of octomeric PAR (~8-mer PAR) binding proteins but not among ~40-mer PAR binders, using photoaffinity probes of defined lengths (Dasovich et al., 2020). As TNKS synthesizes short chain PAR, it is not surprising that VCL has been recovered among ~8-mer PAR binders.

ICF confocal microscopy images allowed to compare VCL in focal adhesions vs. VCL in the adhesion belt, to observe that partial colocalization existed among belt VCL and PAR but there was no PAR signal detected around the VCL located at focal adhesions (Figs. 6C–6Y, Fig. S3, Fig. S4). Thus, the VCL pool PARylated by TNKS (Fig. 6A & Fig. S2) seemed to correspond to the VCL pool bound to the adhesion belt (Figs. 6C–6Y, Fig. S3, Fig. S4).

Then, we further investigated the role of TBM in VCL on epithelial cells.

To be able to compare cell shape changes, a Cell Shape index was designed and validated using an EMT model. CShin proved to be a robust cell shape quantification method and coherent results were obtained. The comparison of graphs in Fig. 1 (NMuMG and EMT), Fig. 5 (TNKSi effects), Fig. 9 and Fig. S8 (“KO” and GgVCL vs. GgVCL/*TBM effects) indicates that in all cases epithelial cells displayed a median CShin <1. It was also evidenced that CShin increased similarly under treatment by TNKSi, VCL “KO” or VCL/*TBM, indicating that any interference on TNKS and VCL interaction had effects on cell shape similar to those observed in the EMT model.

VCL directed mutagenesis and overexpression of Tol2-GgVCL or Tol2-VCL/*TBM in MCF-7 human epithelial breast cancer cells showed that VCL/*TBM induced mesenchymal-like changes in cell morphology. Such changes were more similar to those expected under knockdown conditions than under VCL overexpression, suggesting that the TBM could be crucial to VCL function in cell junctions (Fig. 8, Fig. 9, Fig. S8).

Our transfections with Tol2-GgVCL or Tol2-GgVLC/*TBM had important design limitations that could be solved using a different approach. First, it was meant to achieve transient exogenous GgVCL expression, making it impossible to follow up cells in migration, invasion or chemo-resistance studies. Second, even though we chose a medium-expression promoter (β-actin promoter rather than the stronger CMV), there was a clear GgVCL overexpression. It is worth trying a direct substitution of nucleotides on the endogenous VCL (to obtain G454V aminoacidic change), so that the endogenous VCL gene expression remains unchanged. A long-term expression of TBM*/hVLC, without a change in basal expression level could be achieved through CRISPR-directed edition of the endogenous hVCL gene in MCF-7 cells. In spite of the fact that VCL harbors at least 3 TBMs, we have observed promising interesting changes in cell shape introducing a single aminoacidic substitution in TBM-II.

Conclusions

In this work, we started to test the hypothesis that TNKS PARylates epithelial belt VCL, affecting adherens junctions VCL-dependent functions. To sum up, the hypothesis has several predictions which have been corroborated: (1) VCL harbored TBMs conserved in evolution correlating with adherens junctions functions; (2) TNKS inhibitors disrupted the PAR belt while PAR and a TNKS pool was detected in the plasma membrane fraction: (3) a VCL pool was covalently PARylated in the absence of TNKSi; (4) overexpression of VCL/*TBM induced changes in cell morphology remnant of an EMT, which have been described in the context of VCL knockdown (Rodríguez Fernández et al., 1993) or VCL loss (Li et al., 2014).

The last important prediction of our hypothesis would be that (5) the effect of long-term expression of VCL/*TBM at physiological levels on cell junctions would be EMT induction, implying changes on epithelial morphology, migration and invasion capacity as well as chemo-resistance. This last prediction remains to be tested. Importantly, as the adherens junctions have distinct distribution and roles in certain non-epithelial tissues (Lafon Hughes et al., 2017; Lui et al., 2003), any discovery regarding adherens junctions composition and functions is putatively exciting in fields that look as distant as cancer, infections, neurodegeneration or spermatogenesis. Several processes in different cell types could be modulated by TBM-dependent VCL PARylation by TNKS. Thus, we hope this work will encourage further research in diverse areas.

Supplemental Information

Supplemental Information 1 Supplemental Materials and Methods.

Click here for additional data file.

Supplemental Information 2 Anti-TNKS antibody controls.

Left column: controls without primary antibody. Right column: anti-TNKS ICF was performed with different antibodies (GeneTex rabbit anti-TNKS-Nt or Novex mouse anti-hTNKS, catalytic region (1106-1325)) and under different fixation protocols, namely 4 % PFA or a combination of PFA and glutaraldehyde (bottom).

Click here for additional data file.

Supplemental Information 3 VCL affinity precipitation with Macro resin.

Affinity precipitation of PARylated proteins with Macro resin and WB to detect VCL and PARP-1. (A) Cells were seeded in the absence or presence of TNKSi G007LK and lysed 5 h later. (B) Cells were grown until confluency without or with TNKS+PARP-1/2/3 inhibitor XAV939 and lysed. Arrows point at the VCL species that was enriched in the AP fraction in the absence of TNKSi (next to white asterisks).

Click here for additional data file.

Supplemental Information 4 Colocalization of PAR and VCL in the epithelial belt and colocalization loss in a EMT model.

VCL (red), PAR (green) and colocalizing pixels (yellow). (A-C) Vero cells, 100x; (D): Vero cell monolayer overview; (E- H): correspondent masks; (I - L): masks products highlighting colocalization at the epithelial belt. (M, N): NMuMG cells. (M): untreated or (N): after EMT induction by TGF-β

Click here for additional data file.

Supplemental Information 5 FRET. VCL and PAR colocalization within 50 nm.

FRET was done after indirect ICF using secondary antibodies bound to Alexa 488 as donor fluorophores and secondary antibodies bound to Alexa 546 as receptors. (A – F) PAR -bound Alexa 488 and VCL- bound Alexa 546 and (G - L) viceversa. Then, FRET was evaluated using the ImageJ Fret and Colocalization.plugin, allowing the localization of the subcellular structures where vinculin and PAR colocalized within about 50 nm resolution (10 nm FRET + 10 nm each primary and each secondary antibody, see König et al 2006) (Kwan et al., 2007). (A, G) Relative intensity graphs. (B, H) Donor bleedthrough or spillover image. Red points are more intense than expected by spillover coefficient, will be undercorrected and may give false positive FRET. Thus, it is important to check that the structures of interest are not in red. The opposite is true for blue points. (C, I) Analogous acceptor bleedthrough (D, J) Sample confocal images (E, K) FRET index represents the intensities of the acceptor emission due to FRET. Blue is no FRET, violet/red is FRET (F, L) False positive points with FRET signal in the absence of colocalized donor and acceptor can be excluded, giving this Colocalized FRET index images. Again, violet points represent FRET.

Click here for additional data file.

Supplemental Information 6 MCF-7/”Knock-out” cells: gRNA designs and phenotype (days-1 month).

(A) Each synthetic single-guide RNA (sgRNA) comprises a sequence complementary to the target (CRISPR RNA or guide sequence, here depicted) + a helper and scaffold sequence (transactivating CRISPR RNA) (not-shown). For Cas9 from Streptococcus pyrogenes, the short protospacer adjacent motif (PAM) that has to be downstream the target DNA is 5’-NGG-3’, where N is any nucleotide. The 20 nt-sequences complementary to the gRNA were on the FW strand for gRNA 1, 2 and 4 (yellow rectangles) and on the REV strand for gRNA 3. As all gRNAs were located on the third VCL exon, a primer pair (PP9 FW and PP9 REV) was designed to amplify such region for subsequent checking of the changes obtained for this sequence. (B-E) MCF-7 cells were nucleofected in the absence or presence of RNPs of Cas9 + the indicated sgRNAs and photographed 8 days later. (B) Nucleofection control; (C) sgRNA2 (D, E) sgRNA4. (F-I) MCF-7 cells were electroporated in the absence or presence of RNPs of Cas9 + the indicated sgRNAs and photographed 8 days later (F) Electroporation control; (G) sgRNA1; (H) sgRNA 3; (I) combined sgRNA 1+ sgRNA3. <!--[if !supportLists]-->(B) <!--[endif]--> (J-Q) ICF with anti-VCL antibody (green) and DAPI counterstain (blue) of cells fixed 1 month later. (J, N): electroporation control, (K,O): sgRNA1, (L,P): sgRNA3, (M,O): sgRNa 1+sgRNA3.In electroporation control and sgRNA3, apical and basal regions were still distinguished (the insets represent VCL in basal region). In flattened cells, it was not possible to define apical vs basal regions and there is no inset.

Click here for additional data file.

Supplemental Information 7 MCF-7/”Knock-out” cells phenotype up to 3 months.

Cells electroporated with sgRNA1 + sgRNA3 were followed and subjected to ICF. Here are more examples of microscopic fields: (A-D) one month and (E-H) 3 months after electroporation. These cells transfected with RNPs to knock-out VCL were named MCF-7/”Knock” cells. They were characterized by extremely slow cell cycling, no survival as single cells, cell flattening, and relatively low but variable VCL expression

Click here for additional data file.

Supplemental Information 8 Transfection of “Knocked” cells with either Tol2-GgVCL or Tol2-GgVCL/*TBM.

Cell monolayer was transfected with FUGENE and fixed 48 h later (A-J) or cell suspensions were nucleofected and fixed 20 h later (K-R). GFP (yellow), VCL (green) and F-actin (red) were detected by ICF. (A, F) Monolayer overview, merged channels; (B - E & G-J): enlarged view of two Tol2-VCL/*TBM transfected cells. (B) merge, (C): GFP,(D):VCL+GFP (E): VLC+F-actin+ drawn contour, (G) merge, (H): GFP,(I):VCL+GFP (J): VLC+F-actin+ drawn contour. (K - R) Successful transfection of neighbor cells. (K - N): with Tol2-GgVCL/*TBM; (O - R): with Tol2-GgVCL. (K, O): F-actin, (L, P): VCL, (M,Q): GFP, (N,R): DAPI. Bar: 25 µm

Click here for additional data file.

Supplemental Information 9 Cell shape was affected by VCL "KO", or GgVCL/*TBM.

Click here for additional data file.

We are grateful to everyone who made this work possible. MCF7 cells were a generous gift from Dr. Archana Dhasarathy and Dr. Sergei Nechaev, University of North Dakota, USA, and were routinely cultured at the Cell Culture Room from the Neurochemistry Department, IIBCE. Vero cells were donated by Dr. Santiago Mirazzo and Dr. Juan Arbiza. AccI was a kind gift from Dr. Fernanda Azpiroz, from Microbiology Laboratory at Science Faculty, UdelaR. Anti-PMCA and anti-GRASP antibodies were respectively donated by Dr. Gerardo Corradi, IQUIFIB-CONICET and Dr. Cynthia He Yingxin, National University of Singapore. Dr. Lari Lehtio gently gave us TNKSi. Dr. Marcelo Comini and Dr. Mariana Bonilla gave us open access to the Lonza Nucleofector and the cell culture room at IPMONT. We are particularly indebted to Dr. José Badano and his team, including Dr. Victoria Prieto and Dr. Paola Lepanto, who opened their laboratory at IPMONT and shared with us their reagents, their molecular biology background and their valuable time. We are also grateful to Synthego who donated us the CRISPR/Cas9 VCL KO kit.

Additional Information and Declarations

Competing Interests

Author Contributions

DNA Deposition

Data Availability

The authors declare that they have no competing interests.

Salomé Vilchez Larrea conceived and designed the experiments, performed the experiments, analyzed the data, prepared figures and/or tables, authored or reviewed drafts of the paper, and approved the final draft.

Wanda Mariela Valsecchi analyzed the data, prepared figures and/or tables, authored or reviewed drafts of the paper, and approved the final draft.

Silvia H. Fernández Villamil conceived and designed the experiments, analyzed the data, authored or reviewed drafts of the paper, and approved the final draft.

Laura I. Lafon Hughes conceived and designed the experiments, performed the experiments, analyzed the data, prepared figures and/or tables, authored or reviewed drafts of the paper, and approved the final draft.

The following information was supplied regarding the deposition of DNA sequences:

The 3 plasmids are available at Addgene:

https://www.addgene.org/browse/article/28215513/.

- pET15b/GVcl 1-1066/*TBM: https://addgene.org/162781/.

- Tol2-GgVCL/WT: https://addgene.org/162787/.

- Tol2-GgVCL/*TBM: https://addgene.org/162790/.

The plasmid sequences are available as Snapgene files. Primer walking had been independently carried by Macrogen on TOl2-VCL/*TBM Vcl gene.

The following information was supplied regarding data availability:

The data is available at figshare: - Lafon hughes, Laura (2021): Plasmid sequences. figshare. Dataset. https://doi.org/10.6084/m9.figshare.13260236.v1.

- Vilchez Larrea, Salome (2021): Images for Figshare.zip. figshare. Figure. https://doi.org/10.6084/m9.figshare.13110374.v1.

- Lafon hughes, Laura (2021): aux FRET_ppt. figshare. Presentation. https://doi.org/10.6084/m9.figshare.12747116.v1.

- Lafon hughes, Laura (2021): recalculo 150819 VCL to PAR. figshare. Dataset. https://doi.org/10.6084/m9.figshare.12747104.

- Lafon hughes, Laura (2021): recalculo 150819 PAR to VCL. figshare. Dataset. https://doi.org/10.6084/m9.figshare.12747083.

- Lafon hughes, Laura (2021): aux FRET _070815_serie B_procesados 4. figshare. Dataset. https://doi.org/10.6084/m9.figshare.12747071.v1.

- Lafon hughes, Laura (2021): aux FRET_070815_serie B procesados 3. figshare. Dataset. https://doi.org/10.6084/m9.figshare.12747065.

- Lafon hughes, Laura (2021): aux FRET_070815 serie B_originales OLYMPUS. figshare. Dataset. https://doi.org/10.6084/m9.figshare.12747062.v1.

- Lafon hughes, Laura (2021): aux FRET 070815 serie A_procesados 4. figshare. Dataset. https://doi.org/10.6084/m9.figshare.12747056.v1.

- Lafon hughes, Laura (2021): aux FRET 070815_serie A_procesados 3. figshare. Dataset. https://doi.org/10.6084/m9.figshare.12747053.v1.

- Lafon hughes, Laura (2021): aux FRET_070815 serie A_ORIGINALES 3y4. figshare. Dataset. https://doi.org/10.6084/m9.figshare.12747047.v1.

- Lafon hughes, Laura (2021): aux FRET_registro 060815 serie B_procesados. figshare. Dataset. https://doi.org/10.6084/m9.figshare.12747041.v1.

- Lafon hughes, Laura (2021): aux FRET_060815 serie B_originales OLYMPUS. figshare. Dataset. https://doi.org/10.6084/m9.figshare.12747008.v1.

- Lafon hughes, Laura (2021): aux FRET_registro 060815 serie A_process. figshare. Dataset. https://doi.org/10.6084/m9.figshare.12747005.v1.

- Lafon hughes, Laura (2021): aux Fig FRET_registro 060815 serie A_originales OLYMPUS. figshare. Dataset. https://doi.org/10.6084/m9.figshare.12746978.v1.

- Lafon hughes, Laura; Vilchez Larrea, Salome (2021): aux Fig 1. figshare. Figure. https://doi.org/10.6084/m9.figshare.12746888.v1.

- Lafon hughes, Laura; Vilchez Larrea, Salome (2021): aux Fig 2. figshare. Figure. https://doi.org/10.6084/m9.figshare.12746900.v1.

- Lafon hughes, Laura; Vilchez Larrea, Salome (2021): aux Fig 3. figshare. Figure. https://doi.org/10.6084/m9.figshare.12746906.v1.

- Lafon hughes, Laura; Vilchez Larrea, Salome (2021): supl Figures, old version. figshare. Figure. https://doi.org/10.6084/m9.figshare.12746912.v1.

- Lafon hughes, Laura; Vilchez Larrea, Salome (2021): aux_ Fig S5 _MCF-7/Knock. figshare. Figure. https://doi.org/10.6084/m9.figshare.12746933.v1.

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
