# Peer review of "First body of evidence suggesting a role of a tankyrase-binding motif (TBM) of vinculin (VCL) in epithelial cells"

_PeerJ, doi:10.7717/peerj.11442_

## Round 0.1 · original submission · Major Revisions

· Academic Editor

Major Revisions

As you will see from the reviews, both reviewers and I think this study has merit and we are keen to see this published; however, we all agree that there is a little too much descriptive science here, and some quantification is required in several places throughout the study to support your conclusions. Both reviewers have highlighted areas which require attention. Please address each point in turn and outline your changes in a response letter.

In particular, please pay close attention to the issue raised by reviewer-1 about microscopy images

I look forward to seeing the revised paper in 2021.

Reviewer 1 ·

Basic reporting

English and layout mostly good.
Context of study fairly well described.
Weak written results section: it is very brief and does not sufficiently describe the findings.
No attempts at quantification or statistical analysis of any of the findings, meaning results are purely descriptive.

Experimental design

Comments on specific figures and how they could be improved below:
Fig1: good
Fig2a: Agree that most TNKS is in intracellular membrane pool. There is perhaps a weak band in the plasma membrane fraction. 2b, c and d agree with the fractionation; it looks like nearly all the TNKS is on intracellular membranes. 2d. Some of these arrows don’t seem to point to any discernible staining. Agree that there is faint staining on right-hand cell that could be PM. Would need a plasma membrane marker to confirm this, as well as suitable antibody staining controls (at a minimum secondary antibody alone as well as either: 1. siRNA depletion of TNKS to show specificity of staining. 2. An alternative TNKS antibody showing the same staining pattern. 3. An exogenously expressed, tagged version of the protein also displaying the same localisation).
Fig 3. Are these images all taken with the same microscope settings? The F-actin staining seems to vary markedly between the images, making them difficult to compare. Also, the results text description of figure 3 is extremely brief; simply saying that all the inhibitors effect the PAR belt, and yet the different inhibitor treatments look quite different. Can these be related to the potency/specificity of the inhibitors described in table 2? Also cells in C and H don’t look confluent and therefore probably couldn’t form cell-cell junctions, so authors must ensure seeding density is comparable between all conditions.
Fig 4a good.
Fig4c-s: Evidence for co-localistaion of Par and vincluin not clear, although there may be some overlap. Can co-localisation analaysis on confocal be performed?
Figure 5. Very hard to determine how cells are fitting in monolayer (or otherwise) when neighboring cells not visible. In any case, the two cells expressing mutant vinculin don’t appear any more “irregular” or “mesenchymal” than the cell expressing the wildtype vinculin. I don’t think there is much value to this figure in its current form. Therefore, the inferences made about the function of this site in vinculin are, in my opinion, unjustified by this data.

Validity of the findings

Although based on some potentially interesting observations, the data presented is underdeveloped and, as noted in the first section, nearly entirely descriptive. A major point is that at least some of the data presented here need supporting with quantification. Also, in some cases experiments could fairly easily be extended to make them a lot more informative. For example, in the final figure, the authors have gone to the effort of designing, cloning, expressing and imaging a potentially important mutant version of vinculin and yet the data presented is very basic and therefore cannot support the hypothesis that there may be a functional role of the mutated sequence.

Reviewer 2 ·

Basic reporting

No comment. Meet criteria.

Experimental design

Additional controls, as detailed in "general comments", will make the findings and claims more convincing.

Validity of the findings

no comment.

Additional comments

The authors took a logical approach to support the novel notion that TNKS-related PAR belt near AJ or colocalizes with vinculin at the plasma membrane in VERO cells, and that loss of TNKS-vinculin interaction may promote EMT of MCF cells. Although the designs are well-thought out, some important controls are missing, and some findings are not clearly described.

Fig 2A: ideally, cells should be treated with or without TNKS and PARP1 inhibitors (simultaneously). Bands that disappear upon inhibitor treatment are likely to be bona fide PARylated proteins (not just nonspecific binding to the detecting agents)

Fig. 3. Please use red-channel image (actin alone) to specifical describe how TNKSi alters subcortical actin ring. Does TNKSi make the ring “serrated discontinuous”.

Fig 3: please explain why TNKSi apparently failed to make the PAR belt disappear.

Fig. 3: Highly selective TNKS inhibitors like G007 is not expected to diminish nuclear PAR levels, which are primarily the product of PARP1. Therefore, please explain why nuclear PAR staining intensity is much stronger in panel A than in other panels. Were images acquired and processed using the same parameters?

Fig. 4A The WWE resin, by binding directly to PAR, will precipitate PARsylated proteins as well as (indirectly) their associated partners (which may not be PARsylated per se). This limitation should be included in the discussion. In this figure, it would be very informative (linking vinculin directly to TNKS) to show that after TNKSi treatment, vinculin is no longer detected in the affinity-precipitates.

Fig 4, panels F-K, please use arrow heads to point out yellow pixels, indicative of VCL-PAR colocalization

Fig. 5: please show more cells (not just 1 cell compared against 2 cells) and specifically point out their differences.

Minor concerns:

Title page and line 1: Change “evidences” in the title to “evidence”

Line 79: change loose to loss

Line 121, change to “two being canonical”

Line 260: please elaborate on how MABE 1031 works

---

## Round 0.2 · Minor Revisions

· Academic Editor

Minor Revisions

All agree the work is much improved, and we appreciate your efforts.

A few minor points remain to be clarified/attended to - see reviewer comments. This will not need to go back to the reviewers if these are addressed and clear rebuttal provided.

Reviewer 1 ·

Basic reporting

This has improved a lot. My main concerns were a brief and weakly written results section and lack of quantitation but these have both been clearly addressed.

Experimental design

All my points addressed, as well as a significant new undertaking by the authors to quantify their cell morphology results and improve microscopy images and accompanying figure legends.

Validity of the findings

As mentioned above, findings now backed by significant amounts of quantitation, adding considerably to their validity.

Additional comments

Thanks for making the effort to respond so thoroughly to my initial suggestions. I hope you agree that the manuscript is now significantly improved.

Reviewer 2 ·

Basic reporting

No comments

Experimental design

No comments

Validity of the findings

Line 386: the authors stated that VCL was affinity precipitated by macrodomain resins, and the precipitation was abolished by G007. However, a close inspection of Fig. 2S failed to reveal any VCL in the pulldown. Please clarify. Fig. 2S is critical to the claim that VCL is indeed PARylated by TNKS.

Additional comments

The authors have robustly improved the manuscript. Congratulations.

Minor comments:

Line 73: missing a period
79: loss instead of lose
133: missing a parenthesis
184 “for” 2 hours
196: subjected instead of subject
212: vice versa instead of viceversa
230: remove extra parenthesis
238 typo: symmetry
259: TBM instead of TBD
289: 14.5 instead of 14,5
377: “a fraction of VCL” instead of “a VCL fraction”
396 typo
531: further instead of deeper

table 1: please confirm the concentration unit of DAPI
table 2: please define tcPARP, and reveal the concentration unit for IC50

127: define TBM-II at its first appearance in the text
245: please briefly introduce NMuMG cells
322: please point out the evidence of TNKS being detected in the plasma membrane fraction (as stated in the paragraph heading)
400: please elaborate on this sentence


Several figure legends contain multiple occurrence of mysterious words like <!--[if !supportLists]-->

---

## Round 0.3 · accepted · Accept

· Academic Editor

Accept

Thanks for making these final corrections.